# Maternal Obesity and Offspring Long-Term Infectious Morbidity

**DOI:** 10.3390/jcm8091466

**Published:** 2019-09-14

**Authors:** Gil Gutvirtz, Tamar Wainstock, Daniella Landau, Eyal Sheiner

**Affiliations:** 1Department of Obstetrics and Gynecology, Soroka University Medical Center, Ben-Gurion University of the Negev, Beer-Sheva 8410501, Israel; Sheiner@bgu.ac.il; 2Department of Public Health, Faculty of Health Sciences, Ben-Gurion University of the Negev, Beer-Sheva 8410501, Israel; Wainstoc@bgu.ac.il; 3Department of Neonatology, Soroka University Medical Center, Ben-Gurion University of the Negev, Beer-Sheva 8410501, Israel; DanielleLa@clalit.org.il

**Keywords:** pregnancy, obesity, infections, long term, pediatric hospitalization

## Abstract

Obesity is a leading cause of morbidity world-wide. Maternal obesity is associated with adverse perinatal outcomes. Furthermore, Obesity has been associated with increased susceptibility to infections. The purpose of this study was to evaluate long-term pediatric infectious morbidity of children born to obese mothers. This population-based cohort analysis compared deliveries of obese (maternal pre-pregnancy BMI ≥ 30 kg/m^2^) and non-obese patients at a single tertiary medical center. Hospitalizations of the offspring up to the age of 18 years involving infectious morbidities were evaluated according to a predefined set of ICD-9 codes. A Kaplan–Meier survival curve was used to compare cumulative hospitalization incidence between the groups and Cox proportional hazards model was used to control for possible confounders. 249,840 deliveries were included. Of them, 3399 were children of obese mothers. Hospitalizations involving infectious morbidity were significantly more common in children born to obese mothers compared with non-obese patients (12.5% vs. 11.0%, *p* < 0.01). The Kaplan–Meier survival curve demonstrated a significantly higher cumulative incidence of infectious-related hospitalizations in the obese group (log rank *p* = 0.03). Using the Cox regression model, maternal obesity was found to be an independent risk factor for long-term infectious morbidity of the offspring (adjusted HR = 1.125, 95% CI 1.021–1.238, *p* = 0.017).

## 1. Introduction

Obesity, defined as a body mass index (BMI) of 30 kg/m^2^ or more, is a leading cause of morbidity with increasing prevalence worldwide [1]. Obesity is associated with higher rates of multiple morbidities, including endocrine disease such as diabetes mellitus, cardiovascular disease, certain malignancies, and additionally contributes to all-cause mortality in adults [1]. The prevalence of obesity has been constantly rising, as recent data shows more than third of US adults are considered obese [2]. In this survey from 2007 to 2016, obesity among adults raised from 33% to 39%, and specifically in women, obesity rates soared from 35% to 41% in the last decade. Obesity in women of reproductive age (20–39 years) was 31.8% but increased to 58.5% when overweight (25–30 kg/m2) and obese categories were combined [3].

As one of the most common health problems among women of reproductive age, obesity is of special interest in obstetrics. It disrupts female reproductive processes causing more infertility and early pregnancy losses compared to lean women [4]. In pregnancy, obesity is a known risk factor for an array of adverse maternal, fetal and offspring outcomes [5]. These include pregnancy complications such as gestational diabetes mellitus [6] and hypertensive disorders of pregnancy [7], which may also increase the risk for medically indicated preterm deliveries. Intrapartum complications, including higher rate of failure to progress in labor and elevated risk for cesarean delivery [8]. Postpartum complications such as venous thromboembolism and cesarean scar wound infection are also more likely to occur in obese women [9,10]. In the long-term, obesity during pregnancy was found to be associated with future maternal cardiovascular morbidity [11], ophthalmic complications [12], and even malignancies such as ovarian and breast cancer [13,14].

As for the fetus, maternal obesity has been associated with higher risk for congenital anomalies [15], excessive growth with increased risk for macrosomia [16], and even perinatal death [17]. Being born to an obese mother has short and long-term consequences for the offspring, such as low Apgar score, need for resuscitation at birth and post-partum hypoglycemia [18], and later in life, maternal obesity has been associated with childhood obesity, diabetes mellitus and metabolic syndrome, and possibly an increased risk of cardiovascular morbidity for these children [19].

Obesity also affects human immune response leading to an increased susceptibility to infections. Studies report that obese adults and children show an increased incidence of both nosocomial and community-acquired infections [20]. It has been previously suggested that the in-utero environment in pregnancies affected by malnutrition promotes fetal adaptations via metabolic, hormonal, epigenetic, and placental mechanisms that are the cause of future metabolic morbidity in the offspring [21]. It is possible that obesity during pregnancy also promotes an in-utero environment that will affect the fetal future response to infections. Hence, in this study, we sought to evaluate whether maternal obesity in pregnancy influences offspring susceptibility to pediatric infectious morbidity.

## 2. Methods

In this population based retrospective cohort study, offspring long-term infectious morbidity related hospitalizations were compared among children born to obese and non-obese mothers. The study includes singleton deliveries from a single regional tertiary medical center (Soroka University Medical Center, SUMC). In Israel a prenatal care visits, delivery and offspring hospitalizations are fully covered by national health insurance law, and citizens of all background have access to treatment, therefore our study population is non-selective. The study included deliveries occurring in the hospital between 1991 and 2014, a 23-year period that enables long-term follow-up of the offspring.

Obesity, the independent variable, was defined as a BMI ≥ 30 kg/m^2^, based on height and weight measurement recorded at the first prenatal visit. Deliveries with no BMI record were considered as non-obese. We excluded multiple pregnancies, fetuses with chromosomal or congenital abnormalities and perinatal mortality cases.

Two databases were merged using patients’ ID number (Maternal ID and that given for the offspring at birth): From the obstetric and pediatric departments, to create the study’s dataset. The obstetric database includes maternal and perinatal information recorded at admission and immediately after delivery by an obstetrician. The pediatric consists of demographic information and ICD-9 codes for all medical diagnoses made during hospitalizations of children in the SUMC pediatric departments). The data entered into the datasets is routinely reviewed by medical secretaries assuring coding and diagnoses are complete and accurate.

Using the obstetric database, we analyzed the maternal characteristics and perinatal outcomes including preterm delivery (<37 weeks’ gestation), low birthweight (defined as birthweight < 2500 g), macrosomia (defined as birthweight > 4000 g), cesarean delivery rates and Apgar scores. For long-term outcomes, we included child hospitalizations up to the age of 18 years involving any infectious morbidity. We used a pre-defined set of ICD-9 codes for infectious conditions that are detailed in the Appendix A. Follow-up was terminated for any of the following reasons: at first hospitalization involving infectious morbidity, end of the study period, when the child reached 18 years of age, or child mortality unrelated to infectious morbidity.

### Statistical Analysis

SPSS package 23rd ed. (IBM/SPSS, Chicago, IL, USA) software was used for statistical analysis. Categorical data are shown in counts and rates and the differences were assessed by chi-square tests. Student *t*-test was used for comparison of continuous variables with normal distribution. Kaplan–Meier survival curves were used to compare cumulative hospitalization incidences over time among the study groups. The differences between the curves were assessed using the log-rank test. Only the first admission with any infectious-related condition for a given individual was included in the survival analysis.

A cox regression model was constructed to establish an independent association between maternal obesity in pregnancy and future incidence of infectious-related hospitalizations of the offspring while adjusting for confounding and clinically significant variables, such as: Maternal age, maternal hypertensive disorders of pregnancy (chronic hypertension, gestational or preeclampsia with or without severe features and eclampsia), maternal diabetes mellitus (pre-gestational and gestational), chorioamnionitis during labor, preterm delivery, birthweight, lack of prenatal care, and mode of delivery. All analyses were two-sided, and a *p*-value of ≤ 0.05 was considered statistically significant.

## 3. Results

During the years of the study, 249,840 deliveries in SUMC met the inclusion criteria, of which 3399 (1.4%) were of obese mothers. There were 11,454 multiple gestations (4.3%) and 6105 cases of congenital malformations/chromosomal abnormalities (2.3%) that were excluded from the general cohort. In addition, 1990 (0.8%) perinatal mortality cases were excluded from the long-term analysis. Table 1 summarizes selective maternal characteristics and pregnancy outcomes for both groups. Obese mothers were older, had recurrent pregnancy losses and were more likely to undergo fertility treatments. Pre-gestational diabetes mellitus and hypertensive disorders (chronic, gestational, or preeclampsia) were also more prevalent in this group. In addition, induction of labor was significantly more common in obese mothers and cesarean delivery rates were significantly higher as compared to non-obese mothers. Mean birthweight was higher in the obese group and more children were born with birthweight above 4 Kg. In accordance, there were lower rates of LBW infants in the obese group, compared to the non-obese group. Children of obese mothers had lower Apgar scores at 1 min compared to children of non-obese mothers. Incidence of preterm delivery, chorioamnionitis during labor and perinatal mortality was comparable between groups.

Table 2 summarizes the selected long-term infectious morbidities and hospitalization rates of children up to 18 years of age. Children born to obese mothers had significantly higher rates of respiratory, Otorhinolaryngological (ENT), ophthalmic and viral infections, while other infectious morbidities were comparable. Total infectious-related hospitalization rate was significantly higher for children born to obese patients (12.5% vs. 11.0%, OR = 1.158, 95% CI 1.046–1.283, *p* < 0.01).

In the Kaplan–Meier survival curve (Figure 1), children born to obese mothers had a significantly higher cumulative incidence of hospitalizations involving infectious morbidity, as compared with children born to non-obese mothers (log rank *p* = 0.03).

The Cox regression model used for the association between long-term risk for infectious-related hospitalization in children (up to the age of 18 years) and maternal obesity in pregnancy is presented in Table 3. The model was adjusted for maternal age, hypertensive disorders (chronic hypertension, gestational hypertension or preeclampsia with or without severe features and eclampsia), diabetes mellitus (pre-gestational and gestational), chorioamnionitis during labor, preterm delivery, birthweight, induction of labor, lack of prenatal care and mode of delivery. As compared to non-obese mothers, maternal obesity during pregnancy exhibited a significant and independent association with the long-term risk for infectious-related hospitalization of the offspring with an adjusted hazard ratio of 1.125 (95% CI 1.021–1.238, *p* = 0.017).

## 4. Discussion

This research was meant to extend our knowledge on the long-term consequences of maternal obesity during pregnancy for the offspring. We found that children born to obese mothers had increased risk for pediatric infectious morbidity such as respiratory and viral infections, ENT and ophthalmic infections. Likewise, all other infectious categories that were assessed in this study had a clear trend towards higher rates in the maternal obesity group although those did not reach a statistical significance, probably due to small numbers of hospitalized children. The children of obese mothers were also at increased risk for infectious-related hospitalizations compared to children born to normal weight mothers.

Maternal obesity has many known short and long-term consequences for the fetus such as neonatal respiratory complications and long-term morbidities including child obesity, diabetes mellitus, cardiovascular disease, and even neurodevelopmental delay [22]. This study found a link between maternal obesity in pregnancy and future pediatric infectious morbidity.

The infectious morbidity found in children in our cohort ranged from respiratory illness to ENT and even ophthalmic infections. Some of these associations were also found in previous studies. For example, Haberg et al. [23] were probably the first to report a link between maternal obesity in pregnancy and the development of respiratory illness in children beyond the neonatal period. They found maternal obesity to be related to respiratory infections and wheeze, but when including both background variables and obstetric problems, neither lower respiratory tract infections (LRTIs) nor hospitalizations for LRTIs were associated with high maternal BMI.

Parsons et al. found that elevated maternal pre-gravid BMI was associated with higher risk of early childhood respiratory hospitalizations [24], but their study focused mainly on asthma and other pulmonary conditions, with only some having an infectious etiology. An Australian prospective birth cohort study investigated the association between maternal pre-gravid BMI and offspring all-cause hospitalizations in the first 5 years of life. They found that children of obese mothers had increased risk of all-cause hospital admissions, with respiratory conditions being most excessive. In their study, hospitalizations due to infectious morbidity was also more prevalent in children of obese mothers [25].

A review from 2016 by McGillick and associates [26] describes how maternal obesity affects fetal intrauterine environment to cause altered regulation on fetal lung development leading to downstream risk for respiratory complications at birth and in later life. They explain how factors such as hyperglycemia, hyperinsulinemia, placental transport, and inflammation are altered in the obesogenic intrauterine environment which restrict fetal ability to make a successful transition to the air-breathing environment at birth that might lead to future respiratory morbidity. There is also increasing evidence that inflammation in utero is a potent modulator of lung development, leading to disruption of alveolarization and microvascular development [27]. In light of these findings, the increased respiratory infectious morbidity found in our study could be explained. While this might be true for respiratory illness, an explanation for immunologic pathways in the intrauterine environment causing increased susceptibility to infections is still lacking. Studies show that placentas of obese women exhibit alterations in immune cell populations compared to placentas of non-obese women [28], but these were aimed to explain the inflammatory state leading to metabolic changes in the offspring and not the infectious outcome. Animal studies did found maternal high-fat diet during gestation to affect offspring immune development, resulting in worse outcomes in models of infection, autoimmunity, and allergic sensitization [29], but to date, there are no human clinical trials to suggest an association for future infectious susceptibility.

An association between obesity and infections has been documented in systematic reviews [30,31], focusing on clinical aspects and describe how obese people are more likely than people of normal weight to develop infections of various types including postoperative infections and other nosocomial infections, as well as to develop serious complications of common infections.

Research on obese humans and with obese animal models have repeatedly demonstrated impaired immune function, including decreased cytokine production, decreased response to antigen/mitogen stimulation, reduced macrophage and dendritic cell function, and natural killer cell impairment [32,33]. In a state of low-grade, chronic inflammation that characterizes obesity, the immune system cells and adipocytes have similarities in structure and function [34]. Adipose tissue mediates immune system by the secretion of adipokines, for example, leptin. This specific adipokine, strongly related to obesity, has been extensively studied since it is involved in the regulation of multiple aspects of maternal metabolic homeostasis. Leptin is emerging as a key component in obese pregnancy-related pathologies such as gestational diabetes mellitus [35], and macrosomia [36].

Among its many roles during placentation and embryonic development, the immunomodulatory role of leptin begins in adipose tissue, acting as a pro-inflammatory factor on monocytes and lymphocytes by inducing Th1-type cytokine production [37]. This cytokine, that links metabolic alterations and inflammation, violates the well-balanced system of adipocytes and immune cells, with subsequent disturbance to the immune surveillance system [34]. It has been shown that children born to obese mothers had higher cord blood leptin and adiponectin concentrations than children born to normal weight mothers [38]. Hence, it is possible that these alterations of the immune system in the obese patient, leads to similar immunomodulation in the evolving fetus and thus exposes the offspring to infectious vulnerability later in life.

The obese mothers in our study had higher rates of diabetes mellitus (either pre-gestational or gestational) and hypertensive disorders, co-morbidities that are known to effect obese patients. These co-morbidities have also been linked with adverse pregnancy outcomes and might also have a long-term effect on the offspring. For this reason, we included them as possible confounders in the Cox regression model. Moreover, children of obese mothers are at higher risk to be born by cesarean deliveries, which just recently was associated with increased offspring long-term infectious morbidity [39], so this mode of delivery was also accounted for in the model. Nevertheless, the Cox model, adjusted for these and other possible confounding factors and taking into account the length of follow up, found that maternal obesity during pregnancy was an independent risk factor for long-term infectious morbidity of their children and adolescence.

We believe the major strength of our study is manifested through the statistical power. The ability to collect non-selective population-based data constitutes the study’s main strength. Our success to combine databases from obstetrical and pediatric departments enables us to follow the children born in our hospital all the way to adulthood. Since the Negev region is characterized with positive immigration rate, and SUMC is essentially the sole tertiary medical center in the region, it is expected that children born in our institution will later be hospitalized here, if needed. The combination of these two factors generates a large scale cohort that allows analysis of an adequate sample size comparison.

We recognize that the main limitation of our study is its retrospective design. As such, it merely suggests an association between exposure (maternal obesity) and outcome (child morbidity) and not necessarily causation of the two. Another limitation of this study relates to women with poor prenatal care. The BMI of these women might not have been registered as they did not present, or had late presentation to prenatal exams. Accordingly, since no BMI was registered, they were considered as non-obese and added to the control group.

The incidence of pre-pregnancy maternal obesity in our cohort is very low in comparison to the reported incidence world-wide. However, data published in 2011 found obesity rate in Israel to be extremely lower than OECD average of 17% [40]. Some national surveys found obesity rates in women to be somewhere between 11%–15%, and also importantly, childhood obesity is still considered low at 5.7% [41]. Also to note, the reported prevalence relates to recent years while our study begins in 1991 when obesity rates were even less pronounced. In addition, in our cohort as explained earlier, when patient BMI was not recorded during prenatal care, delivery was counted as non-obese (i.e., comparison group), which can also explain the low incidence. This fact actually emphasizes that the actual association between obesity and long-term risk for infectious morbidity may be even higher since many obese cases were actually considered as non-obese. Thus, although considered a limitation of the study, it is reasonable to believe that our findings are actually underestimating the true association between maternal obesity and the infectious morbidity of their children.

It has been previously shown in extensive literature [42] that increasing BMI leads to increased morbidity, hence we believe that populations with higher obesity would be at an increased risk for adverse pregnancy outcome effecting offspring health, including infectious morbidity.

Another limitation is our ability to include only hospital encounters and not cases that either did not necessitate hospitalization or were diagnosed in an ambulatory setting. It is likely to assume that many infectious illnesses in children are handled in non-hospital encounters, and only the ones that are more severe or complicated by co-morbidity, require hospitalization of the child. Nevertheless, between those cases that were hospitalized, the association between maternal obesity in pregnancy and the child’s infectious morbidity was seen. Finally, as a result of the study’s design, we could not include some demographic characteristics such as social, economic, and residential aspects of the mother and family of the offspring that might contribute to infectious morbidity.

## 5. Conclusions

The findings of this study expands our knowledge on the impact of maternal obesity on future child health and reinforces current recommendations for optimal weight control that should begin before conception. Obese women who have even small weight reductions before pregnancy may have improved pregnancy outcomes [43]. This highlights the importance of pre-conception assessment and screening for obesity to reduce maternal and fetal complications, and a strong recommendation for BMI control should be given routinely to women who are pregnant or planning to conceive. The obstetrician should knowledgably counsel the obese patient on the short and long-term complications of maternal obesity on the neonate, and infectious morbidity possibly being one of them. We believe further studies on this topic will help better understand how obesity in pregnancy is related to offspring immunity and health, and might also highlight pediatric issues that originate in fetal life.

## Figures and Tables

**Figure 1 jcm-08-01466-f001:**
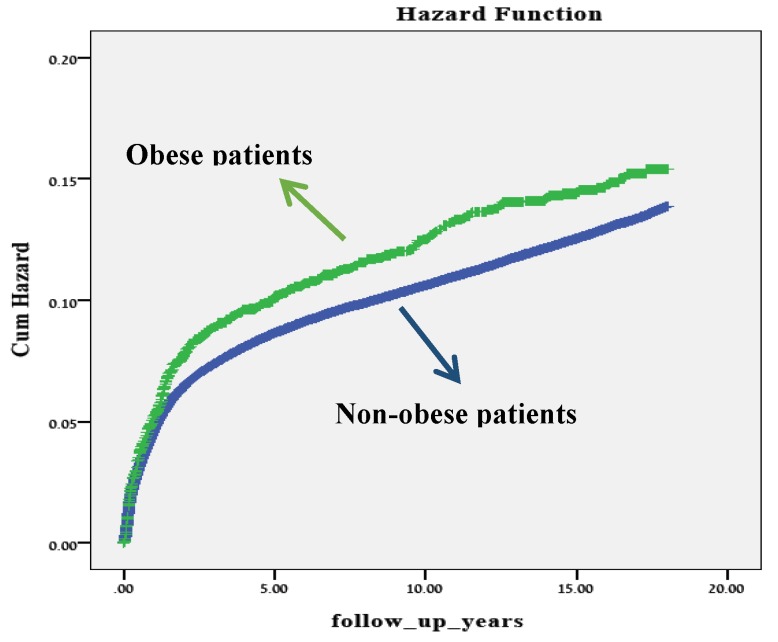
A Kaplan–Meier survival curve showing the cumulative incidence of hospitalizations with any infectious morbidity in children to obese and non-obese mothers.

**Table 1 jcm-08-01466-t001:** Maternal characteristics and pregnancy outcomes of obese and non-obese mothers.

Maternal Characteristic/Pregnancy Outcome	Maternal Obesity *n* = 3399	No Obesity *n* = 246,441	*p* Value
Maternal age (years, mean± SD)	30.4 ± 5.6	28.1 ± 5.8	<0.01
Recurrent pregnancy loss (*n*)	234 (6.9%)	12,417 (5.0%)	<0.001
Fertility treatments ^a^ (*n*)	156 (4.6%)	4340 (1.8%)	<0.001
Diabetes (*n*)	285 (8.4%)	12,293 (5.0%)	<0.001
Pre-gestational	119 (3.5%)	1733 (0.7%)	<0.001
Gestational	166 (4.9%)	10,560 (4.3%)	0.087
Hypertensive disease (*n*)	314 (9.2%)	12,385 (5.0%)	<0.001
Pre-gestational (chronic HTN)	137 (4.0%)	3313 (1.3%)	<0.001
Gestational, preeclampsia and eclampsia	203 * (6.0%)	9755 * (4.0%)	<0.001
Birthweight (gr., mean *±* SD)	3354.1 ± 550	3196.8 ± 521	<0.001
Gestational age at birth (weeks, mean *±* SD)	39.1 ± 2.1	39.1 ± 2.1	0.64
Chorioamnionitis (*n*)	28 (0.8%)	1682 (0.7%)	0.32
Induction of labor (*n*)	1,157 (34.0%)	64,220 (26.1%)	<0.001
Cesarean delivery (*n*)	1,167 (34.3%)	33,189 (13.5%)	<0.001
Preterm delivery (*n*)	244 (7.2%)	17,764 (7.2%)	0.95
34–37 weeks’ gestation	198 (5.8%)	13,784 (5.6%)	0.403
28–34 weeks’ gestation	35 (1.0%)	3098 (1.2%)	0.403
<28 weeks’ gestation	11 (0.4%)	882 (0.4%)	0.403
Low Apgar ^b^ at 1 min. (*n*)	260 (7.6%)	13,800 (5.6%)	<0.001
Low Apgar ^b^ at 5 min. (*n*)	63 (1.9%)	605 (2.5%)	0.02
Low birth weight ^c^ (*n*)	170 (5.0%)	17,612 (7.1%)	<0.001
Macrosomia ^d^ (*n*)	336 (9.9%)	11,344 (4.6%)	<0.001
Perinatal mortality (*n*)	30 (0.9%)	1960 (0.8%)	0.57
Antepartum death	17 (0.5%)	937 (1.4%)	0.262
Intra-partum death	3 (0.1%)	94 (0.1%)	0.141
Post-partum death	10 (0.3%)	929 (0.4%)	0.434

^a^ including all artificial reproductive techniques; ^b^ Low Apgar < 7; ^c^ Low birth weight (LBW) < 2500 g ^d^ Macrosomia > 4000 g; ^*^ Some of the women in this group were diagnosed with superimposed PET; HTN, Hypertension.

**Table 2 jcm-08-01466-t002:** Long-term infectious morbidities in children to obese and non-obese mothers.

Infectious Morbidity	Maternal Obesity *n* = 3399	No Obesity *n* = 246,441	*p* Value
Respiratory infections	214 (6.3%)	13,538 (5.5%)	0.04
Viral infections	42 (1.2%)	2107 (0.9%)	0.01
ENT infections	73 (2.1%)	3616 (1.5%)	<0.01
Ophthalmic infections	17 (0.5%)	727 (0.3%)	0.03
Neonatal infections	8 (0.2%)	679 (0.3%)	0.65
Bacterial infections	7 (0.2%)	342 (0.1%)	0.29
Bacteremia/Septicemia	5 (0.1%)	190 (0.1%)	0.14
Central nervous system infections	6 (0.2%)	564 (0.2%)	0.52
Gastrointestinal infections	62 (1.8%)	4043 (1.6%)	0.40
Total infectious hospitalization	426 (12.5%)	27,133 (11.0%)	<0.01

**Table 3 jcm-08-01466-t003:** Multivariable analysis used for the association between maternal obesity during pregnancy and long-term risk for infectious-related hospitalizations.

Variables	Adjusted HR	95%CI		*p* Value
		Min	Max	
Maternal obesity	1.125	1.021	1.238	0.017
Maternal age (years)	0.998	0.995	1.000	0.027
Birthweight (grams)	1.000	1.000	1.000	1.000
Chorioamnionitis	0.735	0.634	0.853	<0.001
Preterm delivery (<37 weeks)	1.203	1.147	1.262	<0.001
Induction of labor	1.145	1.114	1.177	<0.001
Cesarean delivery	1.220	1.179	1.262	<0.001
Pre-gestational (chronic HTN)	0.865	0.782	0.958	0.005
Gestational, preeclampsia and eclampsia	0.924	0.870	0.982	0.011
Pre-gestational diabetes	1.478	1.314	1.662	<0.001
Gestational diabetes	0.979	0.924	1.039	0.488
Lack of prenatal care	0.750	0.717	0.785	<0.001

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
