# Peer review of "Maternal Obesity and Offspring Long-Term Infectious Morbidity"

_jcm, 2019, doi:10.3390/jcm8091466_

Round 1
Reviewer 1 Report
This retrospective study aimed to study the maternal obesity and the infectious mobility of offspring
major concerns:
As the data presented in table 3: many confounding factors like delivery method. induction or not and preterm delivery also can predict the infectious mobility of offspring (according to the HR, even stronger than obesity). So make the conclusion that maternal obesity links to offspring infectious mobility is not so tight. May be evaluate those cases with obesity without DM or hypertension comparing to cases without obesity also without those medical complications can clear the relationship more clearly. The obesity defined by BMI in this study was obtained in first prenatal visit, do authors exclude cases with hyperemesis gravida or late presentation for antenatal exams ? this would influence the BMI significantly Minor concerns 1. cases be excluded in this study (present as percentage) should be presented.Author Response
major concerns:
As the data presented in table 3: many confounding factors like delivery method. induction or not and preterm delivery also can predict the infectious mobility of offspring (according to the HR, even stronger than obesity). So make the conclusion that maternal obesity links to offspring infectious mobility is not so tight.ANSWER: We thank the reviewer for his comment. The Cox regression model presented in Table 3 was constructed to adjust for the various possible confounding factors that might influence child’s morbidity. Using this model, we found that maternal obesity was an independent risk factor for offspring infectious morbidity taking in to account that these obese patients also have higher rates of co-morbidities such as diabetes mellitus and hypertension.
May be evaluate those cases with obesity without DM or hypertension comparing to cases without obesity also without those medical complications can clear the relationship more clearly.ANSWER: The Cox model presented in Table 3 controlled for diabetes and hypertensive disorders, as well as for other important possible confounders. The association between obesity and long-term risk for infectious-related hospitalizations remained significant.
The obesity defined by BMI in this study was obtained in first prenatal visit, do authors exclude cases with hyperemesis gravida or late presentation for antenatal exams? this would influence the BMI significantlyANSWER: We thank the reviewer for the comment. We did not exclude patients with hyperemesis gravidarum, nor patients with late presentation for antenatal exams. BMI recordings were made at first prenatal care visit. Patients without a BMI recording were considered non-obese. Following the reviewer comment, we added lack of prenatal care to the cox regression model and found it did not change the association between obesity and long-term infectious morbidity of the offspring (HR=1.125, 95%CI 1.021-1.238,p=0.017). This is now highlighted in the limitations of the study.
Minor concerns:
cases be excluded in this study (present as percentage) should be presented.ANSWER: Excluded cases included:
There were 11,454 multiple gestations (4.3%) and 6105 cases of congenital malformations/chromosomal abnormalities (2.3%) that were excluded from the general cohort. In addition, 1990 (0.8%) perinatal mortality cases were excluded from the long-term analysis. This is now presented in the Results section.
Reviewer 2 Report
This paper highlights the relationship between maternal obesity (BMI greater than 30) and long term infectious complictions up to the age of 18.
The manuscript is very interesting, original and well written. In addition it has a significant sample size and adequate statistical analysis.
I have only few minor suggestions for the authors:
It would be interesting to see the BMI distribution of the overweight and normal weight patients with medians and dispersions in order to understand the extent of BMI difference between the two groups. it would be interesting to add infant mortality in the two groups since this obviously detemined interruption of follow up and it is a relevant outcome. How do the authors explain the fact that only 3399 ( 1.4%) of patients of the study group is obese according to their definition? In the introduction they are mentioning data showing rates of obesity in women of reproductive age up to 30%. How do they explain such a low rate of obesity? Is their population particularly virtuous concerning diet and limitation of this problem? What are the implications that can be hyopothesized? What if we were dealing with populations at higher rates of obesity? Do the authors expect higher HRs and risks for the infants in those populations with higher obesity? Please discuss it in the discussion section as well. Table 1. please differentiate between pregestational and gestational diabetes (the authors can leave the cumulative category, but I believe it is useful to provide further information concerning the subgroups). The same should be done for preeclampsia, preterm delivery below 34 weeks and/or 28 weeks. The same approach needs to be considered for table 3, if possible. Discussion and references are adequate. The discussion can be shortened a little bit, without loosing its relavant scientific content (there are many repetitions at lines 241-248) I think that the authors need to conclude in their discussion with a strong statement concerning the need to control of BMI before pregnancy and this should be one of the important target to adress during preconceptional assessment. This needs to be done discussing the risk for the offspring, besides maternal risks which are known to a larger extent in the general population. I think the authors need to make a strong call for further studies on this topic. I also think that the authors need to emphasize the statistical power of their study within the strenght of the study. I congratulate with the authors for their work.Author Response
I have only few minor suggestions for the authors:
It would be interesting to see the BMI distribution of the overweight and normal weight patients with medians and dispersions in order to understand the extent of BMI difference between the two groups.ANSWER: We thank the reviewer for this important comment. Unfortunately, we do not have data on BMI distribution as the exposure variable (Obesity defined as a body mass index (BMI) of 30 kg/m2 or more) was recorded by a dichotomous variable (obesity yes/no) and not BMI distribution.
it would be interesting to add infant mortality in the two groups since this obviously determined interruption of follow up and it is a relevant outcome.ANSWER:According to the comment of the reviewer we have added data on total perinatal mortality as well as antepartum perinatal mortality, intrapartum and post-partum death (Table 1 in the manuscript). Regarding infant mortality, children who were diagnosed with an infectious morbidity were defined as cases (follow up ended with an event), while children who have died unrelated to infectious morbidity during the study period were censored from the analysis.
How do the authors explain the fact that only 3399 (1.4%) of patients of the study group is obese according to their definition? In the introduction they are mentioning data showing rates of obesity in women of reproductive age up to 30%. How do they explain such a low rate of obesity? Is their population particularly virtuous concerning diet and limitation of this problem? What are the implications that can be hypothesized? What if we were dealing with populations at higher rates of obesity? Do the authors expect higher HRs and risks for the infants in those populations with higher obesity? Please discuss it in the discussion section as well.ANSWER: We thank the reviewer for the important comment. The incidence of pre-pregnancy maternal obesity in our cohort is very low in comparison to the reported incidence world-wide. However, data published in 2011 found obesity rate in Israel to be extremely lower than OECD average of 17% (1). Some national surveys found obesity rates in women to be somewhere between 11-15%, and also importantly, childhood obesity is still considered low at 5.7% (2). Also to note, the reported prevalence relates to recent years while our study begins in 1991 when obesity rates were even less pronounced. In addition, in our cohort, when patient BMI was not recorded during prenatal care, delivery was counted as non-obese (i.e. comparison group), which can also explain the low incidence. This fact actually emphasizes that the actual association between obesity and long-term risk for infectious morbidity may be even higher since many obese cases were actually considered as non-obese. Thus, although considered a limitation of the study, it is reasonable to believe that our findings are actually underestimating the true association between maternal obesity and the infectious morbidity of their children.
It has been previously shown in extensive literature (3)that increasing BMI leads to increased morbidity, hence we believe that populations with higher obesity would be at an increased risk for adverse pregnancy outcome effecting offspring health, including infectious morbidity. We have added this paragraph to the discussion section. (see references below, also added to the paper).
Table 1. please differentiate between pre-gestational and gestational diabetes (the authors can leave the cumulative category, but I believe it is useful to provide further information concerning the subgroups).ANSWER: According to the comment we have differentiated between pre-gestational and gestational diabetes (see Table 1 in the manuscript).
The same should be done for preeclampsia, preterm delivery below 34 weeks and/or 28 weeks.ANSWER: According to the comment we have differentiated between chronic hypertension and preeclampsia and also gave data on PTD <37 as well as <34 <28 (see Table 1 in the manuscript).
The same approach needs to be considered for table 3, if possible.ANSWER:According to the comment we have constructed a new Cox regression model (Table 3 in the manuscript) including the division of hypertensive disorders in pregnancy and diabetes mellitus. Regarding preterm delivery, we did not differentiate the gestational age since the distribution between groups is comparable as can be seen in Table 1 (in the manuscript).
Discussion and references are adequate. The discussion can be shortened a little bit, without loosing its relevant scientific content (there are many repetitions at lines 241-248)ANSWER: The discussion was revised according to the reviewer comment.
I think that the authors need to conclude in their discussion with a strong statement concerning the need to control of BMI before pregnancy and this should be one of the important target to address during pre-conceptional assessment. This needs to be done discussing the risk for the offspring, besides maternal risks which are known to a larger extent in the general population.ANSWER: We thank the reviewer for the comment. The discussion was revised accordingly and a strong recommendation for pre-conceptional control of maternal BMI was added.
I think the authors need to make a strong call for further studies on this topic.ANSWER: We certainly agree. We added this comment to the end of the paper, calling for further studies.
I also think that the authors need to emphasize the statistical power of their study within the strength of the study.ANSWER: The discussion was revised accordingly adding the statistical power as a major strength of the study.
I congratulate with the authors for their work.ANSWER: We thank the reviewer for his kind words.
Refernces
OECD. Health at a Glance 2011: OECD Indicators, OECD Publishing. 2011. (MOH) IMoH. Prevention and treatment of obesity 2011 [Available from: https://www.health.gov.il/PublicationsFiles/Obesity-prof_en.pdf. Dobner J, Kaser S. Body mass index and the risk of infection - from underweight to obesity. Clin Microbiol Infect. 2018 Jan;24(1):24-8. PubMed PMID: 28232162. Epub 2017/02/20. eng.Round 2
Reviewer 1 Report
The authors have responded my comments well.
Author Response
We Thank the reviewer for his thoughtful comments and the opportunity to improve our manuscript.
Reviewer 2 Report
I believe that the authors succeeded in improving the manuscript, following most of the suggestions provided in my first revision. I believe that there may be some further improvements if they look at the following points:
Criticisms which are unresolved (such as the dichotomic approach of BMI, etc) should appear in the limitation section of the study. Please add. The abstract should include a sentence of conclusion/comment after the presentation of hazard ratio. I believe that a sentence like : “maternal obesity was shown in this study as a predictor of infectious morbidity in infants up to the age of .... and this may be due to.... (or similar equivalent) the word count of the abstract, introduction, and discussion may be reduced more, without loosing content. the conclusion of the paper is a bit redundant, repetitions need to be avoided. I suggest to add a header in the beginning of the discussion titled main findings with a single sentence summarizing the findings of the study.
Author Response
I believe that the authors succeeded in improving the manuscript, following most of the suggestions provided in my first revision. I believe that there may be some further improvements if they look at the following points:
Criticisms which are unresolved (such as the dichotomic approach of BMI, etc) should appear in the limitation section of the study. Please add.
ANSWER: We have added the BMI recording approach to the methods section and the consequential limitation to the discussion section – limitations of the study.
The abstract should include a sentence of conclusion/comment after the presentation of hazard ratio. I believe that a sentence like : “maternal obesity was shown in this study as a predictor of infectious morbidity in infants up to the age of .... and this may be due to.... (or similar equivalent) the word count of the abstract, introduction, and discussion may be reduced more, without loosing content.ANSWER: The abstract was revised as suggested.
the conclusion of the paper is a bit redundant, repetitions need to be avoided. I suggest to add a header in the beginning of the discussion titled main findings with a single sentence summarizing the findings of the study.ANSWER: We have revised the conclusion as requested by the reviewer and added the Main findings header before the conclusion section.
This manuscript is a resubmission of an earlier submission. The following is a list of the peer review reports and author responses from that submission.